# Autophagy–Lysosome Pathway in Multiple System Atrophy Pathogenesis: The Best Is Yet to Come

**DOI:** 10.3390/ijms262010204

**Published:** 2025-10-20

**Authors:** Panagiota Mavroeidi, Maria Xilouri

**Affiliations:** Center of Clinical Research, Experimental Surgery and Translational Research, Biomedical Research Foundation of the Academy of Athens, 11527 Athens, Greece; panagiotama88@yahoo.gr

**Keywords:** α-Synuclein, autophagy, lysosomes, multiple system atrophy (MSA), neurodegeneration, oligodendrocytes, therapeutics

## Abstract

Multiple lines of evidence extracted from human post-mortem brain material and cellular and animal models of concomitant proteinopathies cumulatively suggest that the neuronal protein α-Synuclein exerts a strong influence on the pathogenesis of neurodegenerative comorbidities, collectively termed α-Synucleinopathies. Accumulation of α-Synuclein-positive inclusions in neurons or oligodendrocytes is the main histopathological hallmark of Parkinson’s disease (PD) or multiple system atrophy (MSA), respectively. In addition, various pieces of data indicate that components of the autophagy-lysosomal pathway are altered in the context of α-Synucleinopathies. α-Synuclein itself is degraded by autophagy; however, aberrant protein conformations may impair lysosomal function. Genetic PD often involves components of the lysosome, including common genetic mutations in *GBA1*, which encodes for the lysosomal enzyme *β*-glucocerebrosidase. Alterations in lysosomal components that correlate with a commensurate increase in α-Synuclein deposition have been widely observed in PD brains. However, corresponding data in the context of MSA are emerging but remain less extensive than PD. In the current review, we focus on the pathological features as well as the impairments in the autophagy–lysosome pathway (ALP) that are associated with MSA and discuss the current challenges and future directions of therapeutic strategies targeting autophagy in experimental MSA-like models.

## 1. Introduction

Multiple system atrophy (MSA) is a rare, fatal, adult-onset neurodegenerative disorder characterized by the progressive loss of the autonomic nervous system, with only symptomatic treatment currently available [1]. MSA, together with Parkinson’s disease (PD) and dementia with Lewy bodies (DLB), is collectively termed α-Synucleinopathies due to the pathological accumulation of the neuronal protein α-Synuclein (αSyn) within the proteinaceous inclusions formed in the cytoplasm of neurons or glial cells [2,3,4]. Specifically, in MSA αSyn, along with the oligodendroglial-specific phosphoprotein tubulin-polymerization promoting protein TPPP/p25α [5], deposit mostly within the cytoplasm of oligodendrocytes, the myelinating cells of the central nervous system (CNS), leading to the composition of glial cytoplasmic inclusions (GCIs) [6,7]. The origin of αSyn found in oligodendrocytes in MSA brains remains ambiguous, as mature oligodendrocytes are considered not to express detectable levels of the protein under baseline conditions [8,9]. However, according to the prevailing hypothesis, αSyn is internalized by oligodendrocytes, most probably following its release by neurons in a free form or associated with extracellular vesicles, thus forming pathological aggregates that spread to neighboring cells in a prion-like manner, due to the protein’s nucleation propensity [10,11,12,13,14] (Figure 1).

Alternatively, αSyn in oligodendrocytes may not only reflect neuronal uptake but also stress-induced endogenous expression, particularly under conditions where proteolytic systems, such as the autophagy–lysosome pathway (ALP) or the ubiquitin–proteasome system (UPS), are impaired (Figure 1). Various studies have proposed that αSyn detected within oligodendrocytes could be the result of enhanced expression and/or decreased degradation of the endogenous protein [15,16]. We and others have shown that indeed rodent oligodendroglial cell lines and primary cultures express low to non-detectable levels of αSyn, which increase upon autophagy or proteasome impairment [14,17,18]. ALP, the main degradation pathway responsible for the clearance of abnormal protein aggregates and organelles within the CNS [19], represents a key factor contributing to the accumulation of aggregate-prone models of neurodegenerative diseases. Data from human post-mortem material suggest that ALP components co-localizing with αSyn are present within LBs and GCIs [20,21]. Most neurological diseases, such as Alzheimer’s disease (AD) [22], PD [23], DLB and MSA [17,18] are linked to ALP failure and subsequent accumulation of disease-related aberrant protein species.

Three distinct types of autophagy have been identified, depending on the route via which cytosolic components are delivered into the lysosomal lumen for degradation [24]: macroautophagy, chaperone-mediated autophagy (CMA), and microautophagy. During the last decade, emerging studies have proposed a significant involvement of macroautophagy in MSA pathogenesis; however, the role of CMA has not been fully elucidated yet. [15,18,25]. Numerous strategies have been applied in order to modulate the levels of key proteins involved in the ALP machinery, including transcriptional and epigenetic mechanisms, as well as post-transcriptional modifications or microRNA application. [26,27]. Recent findings from our lab underpin the implication of both macroautophagy and CMA in the clearance of oligodendroglial αSyn and TPPP/p25α proteins under both physiological and pathological MSA-like conditions. In addition, molecular and pharmacological manipulation of macroautophagy or CMA (down- or up-regulation) seems to affect the aggregation state of these two main GCI components within oligodendrocytes [17]. Such autophagy-targeted strategies are of great research interest since they could have therapeutic utility in MSA and related α-Synucleinopathies. In the current review, we are discussing the involvement of the various proteolytic machineries in the pathogenesis of MSA, focusing mainly on ALP, which seems to hold a central role in the formation and the removal of oligodendroglial αSyn aggregates within oligodendrocytes.

## 2. The ALP in Action

The ALP, consisting as mentioned above of macroautophagy (Figure 2A), CMA (Figure 2B), and microautophagy (Figure 2C), is responsible for the delivery of cellular components, damaged organelles, and protein aggregates to the acidic environment of the lysosomes for subsequent degradation. Macroautophagy is mediated by the formation of double-membrane vesicles, called autophagosomes (autophagic vacuoles, AVs), which are then fused with the lysosomes for bulk degradation [28]. The first step of mammalian macroautophagy includes the formation of the phagophore (induction and nucleation step), which then engulfs the cytoplasmic material, followed by the elongation of the phagophore membrane and the fusion of its edges to finally form the autophagosome (elongation, closure, and maturation step). Then, the autophagosome is delivered to the lysosome to form the autolysosome (fusion step), where the digestive enzymes (hydrolases) are responsible for the degradation of the luminal material (Figure 2A). The resulting materials are then released and recycled in the cytosol. Alternatively, the autophagosome may fuse with an endosome to form an amphisome, before fusing with the lysosome [28].

On the other hand, CMA is a very selective autophagic pathway, responsible for the removal of proteins that contain the KFERQ pentapeptide sequence or other biochemically related motifs [29]. All CMA-related substrates are recognized by the cytosolic chaperone heat shock protein 70 (Hsp70), the co-chaperone heat shock cognate protein 70 (Hsc70) and other co-chaperones, and, following unfolding, are transported to the lysosomal membrane where binding with the lysosome-associated membrane protein type 2A (LAMP2A) takes place [30,31,32,33,34]. LAMP2A is the rate-limiting step of the pathway, acting as a specific receptor of CMA substrate proteins, which are then introduced into the lysosomal lumen for digestion by lysosomal proteases [35] (Figure 2B).

Finally, the lesser-known mechanism of self-eating, termed microautophagy, is a degradation process that involves the direct engulfment of the cytoplasmic cargo via membrane invagination, and the vesicles that are formed are targeted to the lysosomes for protein, lipid, or organelle digestion [36,37] (Figure 2C). Moreover, a process with similar characteristics to yeast microautophagy operates in mammals and in late endosomes/multivesicular bodies instead of lysosomes [38]. This process, termed endosomal microautophagy (eMI) (Figure 2(Ci,Cii)), contributes to the bulk degradation of proteins present in the cytosol that are trapped in vesicles forming at the late endosomal membrane.

## 3. Unraveling the Role of ALP at the Crossroads of MSA Pathophysiology

The role of the intracellular degradation pathways in the development of neurodegenerative disorders has been extensively studied during the last decades. Regarding α-Synucleinopathies, it has been clearly postulated that the impairment of both the ALP and the ubiquitin-proteasome system (UPS) is closely related to the pathogenesis of PD [39,40]. However, little is known about the implication of defective proteostasis in the development of MSA, which is considered a primary oligodendrogliopathy—indicating that pathology originates predominantly within oligodendrocytes—and a secondary neuropathy, implying that neuronal degeneration arises as a downstream consequence of oligodendroglial failure [41]. Herein, we synthesize ALP/UPS evidence in MSA, noting contrasts with PD.

### 3.1. Proteolytic Enzymes and αSyn Degradation

A seminal study by Iwata and colleagues in 2003 highlighted the importance of proper αSyn degradation in MSA pathogenesis. In this work, it was suggested that the serine protease neurosin (kallikrein-6) is responsible for the proteolysis of α-synuclein (αSyn) and that downregulation of this enzyme is associated with αSyn accumulation in the disease state (Figure 3A). Specifically, it was shown that silencing neurosin expression in SH-SY5Y cells using siRNA resulted in elevated levels of full-length αSyn (n = 3). [42]. This hypothesis was supported by their in vivo findings of αSyn and neurosin co-localization at axon terminals in the mouse brain striatum and by the presence of a strong neurosin signal within αSyn+ inclusions found in human MSA brains [42].

Neurosin cleaves αSyn present in both physiological and pathological conformations [42,43,44]. Various studies were focused on the role of neurosin in the pathogenesis of α-Synucleinopathies, with a particular interest in its ability to reduce the levels of αSyn in oligodendrocytes both in vitro and in vivo [42,45,46]. Specifically, Spencer and colleagues suggested that the delivery of genetically modified neurosin with increased half-life (R80Q mutation) in the brain of an MSA mouse model overexpressing human αSyn specifically within oligodendrocytes (MBP-αSyn mice) reduced levels of aggregated αSyn in both oligodendrocytes and astrocytes. Interestingly, the detection of positive αSyn signal within microglial cells allowed the authors to propose the hypothesis that neurosin inhibits the transfer of αSyn from oligodendrocytes to astrocytes, whereas microglia are responsible for the protein’s clearance [45] (Table 1).

Additionally, the finding of increased expression of the three proteases kallikrein-6 (KLK6), calpain-1 (CAPN1), and cathepsin-D (CTSD) in various brain regions of MSA patients, such as the putamen, pontine base, and cerebellar white matter, has been proposed to be a compensatory response to the presence of increased αSyn load [46]; however, the levels of other proteases (i.e., matrix metalloproteinase-3, MMP-3, and endothelin-converting enzyme, ECE) remain to be investigated in the context of MSA [47,48]. Taking into consideration that these proteases are found within LBs and GCIs and are directly related to the regulation of αSyn levels [49,50,51], it has been suggested that the recruitment of these three enzymes may act as a protective mechanism of the cells or as a response to the abnormal accumulation of αSyn under a disease state [46].

### 3.2. Autophagy Dysfunction in MSA

#### 3.2.1. The Contribution of Adaptor Proteins and miRNAs

Another study highlighted dysregulation of upstream autophagy regulators in MSA. Specifically, the authors performed immunostaining analysis using human brains of MSA patients, and they uncovered a relationship between AMBRA1 (autophagy/beclin1 regulator 1) and both native and phosphorylated (pathological) αSyn [52]. AMBRA1 is an adaptor protein that regulates autophagy and is responsible for the efficient degradation of αSyn, which was identified as a component of GCIs, displaying high affinity with aberrant αSyn species [52]. Similarly, in neuronal models, it has been shown that silencing AMBRA1 in mouse primary cultured neurons led to an accumulation of dot-like endogenous αSyn structures, a phenotype that was similarly induced by treatment with bafilomycin, an inhibitor of autophagosome–lysosome fusion [52] (Figure 3B).

Moreover, induction of macroautophagy by the geldanamycin analog 17-AAG [17-(Allylamino)-17-demethoxygeldanamycin] has been shown to reduce the aggregation of αSyn, in contrast to the proteasomal modulation, which had no effect on αSyn levels. Moreover, LC3-II, which, together with SQSTM1/p62, is a widely used marker for autophagic flux, indicative of the presence of autophagosomes, was found punctuated upon treatment of both primary oligodendrocytes and OLN-A53T cell line (oligodendrocytes overexpressing the human mutated αSyn) with 17-AAG (Table 2). On the contrary, addition of the macroautophagy inhibitor 3-methyladenine (3-MA) in OLN-A53T cells evoked the reverse effects on the LC3-II pattern and led to the enlargement of the formed αSyn aggregates (Figure 3C) [53].

Interestingly, the autophagic adapter protein NBR1, encompassing both LC3- and ubiquitin-binding domains, has been recognized as a component of GCIs, suggesting that decreased autophagic activity is linked to aggregate formation in the context of MSA [54] (Table 1). The fact that this protein was absent in other neuronal and glial inclusions in various proteinopathies led the authors to the hypothesis that NBR1 preferentially binds to αSyn-related protein components [54]. In addition, the FBXO47 gene, which encodes a protein associated with ubiquitination and protein degradation, was identified as a potential risk factor for MSA in a genome-wide association study [55].

Furthermore, the levels of miRNAs implicated in the expression of autophagy-related genes were measured in the context of the contribution of transcriptional and epigenetic mechanisms in ALP dysfunction. Interestingly, the microRNAs miR-101 and let-7b were found to be upregulated in the striatum of MSA brains, which suppressed genes participating in the autophagy process, thus leading to increased αSyn accumulation due to autophagy defect in disease states (Figure 3D and Table 2) [56].

**Table 2 ijms-26-10204-t002:** Experimental and clinical evidence from autophagy–lysosome–proteostasis–targeting strategies in MSA (→: leads to, **↑**: increase, ↓: decrease).

Strategy/Agent	Primary Target & Intended Effect	Model/System	Reported Outcomes	Translational Status	Reference
KYP-2047 (prolyl oligopeptidase inhibitor)	PP2A activation → ↑Autophagy	OLN-AS7 cells	↓ αSyn+ inclusions ↓ pSer129-αSyn ↑ viability	Preclinical (cellular)	[57]
Rapamycin/Sirolimus	mTOR inhibition → ↑Macroautophagy	Oligodendroglial cells primary oligodendrocytes αSyn PFFs	Removal of pathological αSyn species; trial: no clinical benefit	Clinical trial (MSA): no benefit (48 weeks)	[58]
17-AAG (Hsp90 inhibitor)	Chaperone reprogramming → ↑Macroautophagy	Primary oligodendrocytes OLN-A53T	↑ LC3-II puncta ↓ aggregates effect blocked by 3-MA	Preclinical (cellular)	[53]
Anti-miR approaches (miR-101, let-7b)	De-repress ALP genes → restore flux	Human MSA striatum (↑miR-101/let-7b), cell models	Predicted ↓ αSyn via restored ALP	Preclinical concept	[56]
Neurosin/KLK6 augmentation (R80Q)	Proteolysis of αSyn (extracellular/aggregated)	MBP-αSyn transgenic mouse	↓ Aggregated αSyn in oligodendrocytes/astrocytes ↑ microglial uptake	Preclinical (in vivo)	[45]
UCH-L1 inhibition	DUB modulation → ↑Autophagy	Oligodendroglial cell models	Prevents αSyn aggregate formation via autophagy activation	Preclinical (cellular)	[59]

#### 3.2.2. Aggresome-Related Mechanisms

Another protein-component of αSyn-rich GCIs within oligodendrocytes is SQSTM1/p62. Initial studies in post-mortem brains from patients with combined MSA and AD revealed strong immunoreactivity for SQSTM1/p62, mainly within neurofibrillary tangles (NFTs), which are the neuropathological hallmark of AD, and less frequently in αSyn+ neuronal inclusions [60]. However, co-existence of SQSTM1/p62 and αSyn mainly occurred in glial inclusions rather than neuronal aggregates, thus suggesting that αSyn accumulation is governed by different mechanisms in neurons and glial cells [60]. Moreover, SQSTM1/p62 has been proposed to act as an aggresome-related protein along with ubiquitin, molecular chaperones, dynein motor, histone deacetylase 6 (HDAC6), and proteasome subunits [61,62,63]. Since HDAC6 has been identified as a component of GCIs in MSA together with αSyn [63], it has been proposed that αSyn aggregates in oligodendrocytes are associated with aggresomes [64]. This finding led Miki and his colleagues to hypothesize that the formation of both LBs and GCIs is a protective mechanism of cells, enabling the removal of redundant levels of toxic cytoplasmic proteins (Figure 3E) [64].

It has been additionally proposed that aggresomes are microtubule-related inclusions positive for gamma-tubulin (γ-tub) protein, apart from HDAC6 and proteasomal elements, and this mechanism seems to govern the accumulation of GCIs within oligodendrocytes (Table 1). In fact, it is thought that γ-tub is located at the core of aggresomes and is responsible for the recruitment of misfolded proteins, finally leading to αSyn accumulation in both oligodendrocytes and neurons [65].

#### 3.2.3. Defective Autophagy Maturation

GABARAP is an Atg8 homolog in mammals, which plays a significant role in autophagosome maturation [66]. Importantly, it has been shown that mature GABARAP was downregulated in the cerebellum of MSA patients, whereas the levels of LC3 detected in detergent-resistant fractions were increased (Table 1). Moreover, the staining with anti-LC3, anti-GABARAPs, or anti-GATE-16 (a member of the GABARAP family) antibodies revealed defective autophagy maturation and impaired autophagic flux due to decreased levels of autophagy-related proteins and elevated levels of adaptor proteins, such as SQSTM1/p62, in MSA (Figure 3F) [25].

#### 3.2.4. αSyn and ALP Impairment in MSA Pathogenesis

The literature encompasses controversial results regarding the impact of pathological αSyn conformations on ALP initiation and completion within oligodendrocytes. The involvement of the ALP in the pathogenesis of MSA was initially investigated in 2012, when it was proposed that various lysosomal proteins were found to be increased in the brains of MSA patients. However, the majority of these lysosomal proteins were not co-localized with αSyn+ cells, but, on the contrary, lysosomal alterations were observed in microglia, thus suggesting that these cells are primarily activated, finally leading to an oligodendroglial pathology [67]. According to their hypothesis, αSyn does not lead to direct lysosomal activation in MSA cases; nonetheless, microglial lysosomes become activated and altered as a response during the development of the disease [67] (Table 1). On the other hand, another study has proposed that the augmented levels of LC3-II were associated with aggregated αSyn found within GCIs in the brains of seven MSA cases [15]. Immunostaining analysis using antibodies against LC3-II, SQSTM1/p62, and ubiquitin revealed ALP upregulation under pathological conditions, which was probably attributed either to inhibition of autophagic flux or to impaired proteasomal activity [15].

It has also been proposed that either the overexpression of human αSyn or the incubation of the OLN-t40 cell line with αSyn pre-formed fibrils (PFFs) does not affect the autophagic flux. Although αSyn seems to be mainly degraded via the ALP, the authors support that autophagic flux is disturbed only upon mitochondrial impairment and oxidative stress [68].

In an attempt to further elucidate the role of autophagy in the formation of oligodendroglial αSyn+ inclusions, Fellner and colleagues inhibited macroautophagy and added either monomeric or fibrillar αSyn in human oligodendroglial cells [69]. According to their results, the pharmacological (using bafilomycin) or genetic (knockdown of *LC3B*) inhibition of macroautophagy in the PFF-treated cells was not able to induce the pathological accumulation of αSyn and the formation of GCI-like inclusions within oligodendrocytes, thus suggesting that probably multiple factors lead to the formation of GCIs in MSA [69,70]. The authors conclude that although no further aggregation of pathological αSyn was observed upon macroautophagy inhibition, higher amounts of αSyn or longer incubation times should be applied in order to draw safer conclusions [69].

In another study, the role of the lysosomal response against αSyn aggregation was evaluated in human brains of MSA and PD patients and in healthy controls, using immunohistochemistry analysis of CTSD or double immunolabeling (CTSD/αSyn) laser confocal microscopy. The authors surmised that since different strains of αSyn are responsible for the distinct phenotypes of α-Synucleinopathies, it is possible that the low co-localization of CTSD with pathological conformers of αSyn detected in MSA, in contrast to PD cases, may be due to the high vulnerability of oligodendrocytes in the MSA-related αSyn strains [71].

Furthermore, impaired mitochondrial function and autophagy deficiency seem to be strongly associated with MSA pathogenesis, which was accompanied by dysregulated activity of lysosomal enzymes, as proposed by the study of Compagnoni and colleagues in 2018 [72]. The authors established a humanized in vitro model of MSA using iPSC-derived neurons from patients and healthy controls and assessed the levels of LC3-II, demonstrating a dysregulation in autophagy and a concurrent decrease in CoQ10, thus revealing the presence of mitochondrial defects in MSA neurons [72].

### 3.3. The Contribution of Astrocytes

Similar to oligodendrocytes, αSyn has been reported to be taken up by astrocytes as well. Internalized αSyn has been found to be localized in the astrocytic lysosomal compartments and Rab7-positive structures, which are related to late endosomes. Although it was already known that αSyn can be taken up by astrocytes [73,74], in this work, it was further suggested that astroglial cells are able to engulf oligodendroglial cell debris via phagocytosis [71], thus highlighting the contribution of proper astroglial lysosomal activity in the development and progression of MSA.

### 3.4. TPPP/p25α and Autophagy Inhibition

Beyond αSyn, another major component of GCIs and an inducer of αSyn aggregation is the oligodendroglial phosphoprotein TPPP/p25α [75]. Likewise, the contribution of TPPP/p25α in ALP impairment has also been explored. In an in vitro study, the implication of TPPP/p25α in autophagy dysfunction was evaluated in the well-known PC12 pheochromocytoma cell line overexpressing the human PD-linked mutant A30P αSyn in the presence or absence of TPPP/p25α. According to their results, the overexpression of TPPP/p25α inhibited the fusion of autophagosomes with lysosomes, due to the inhibition of HDAC6 activity, an event that stimulated the secretion of αSyn (both monomeric and high molecular weight species) via exophagy (Figure 3G) [76]. Interestingly, another group obtained similar results regarding the role of TPPP/p25α in the proteolytic degradation of αSyn via the autophagic machinery. In this work, a mixture of recombinant αSyn and TPPP/p25α proteins was added to both HeLa and SHSY-5Y cell lines, and the degradation of αSyn was assessed. Again, TPPP/p25α seemed to hinder the proteolysis of αSyn via the ALP system due to its binding to LC3B, which inhibited the autophagy maturation [77]. The above results were further supported by a recent study from our group, which demonstrated that TPPP/p25α overexpression led initially to macroautophagy inhibition, which was followed by an upregulation of CMA, probably as a compensatory response of oligodendrocytes to counteract the excess αSyn protein load [17].

## 4. Targeting the ALP to Counteract MSA-like Pathology

### 4.1. Macroautophagy in MSA

Although the presence of mRNA or protein levels of the oligodendroglial αSyn has been debated in the literature [8,9,16,78,79], recent findings support that the endogenous protein is indeed expressed within oligodendrocytes, even in minuscule amounts, and that it plays a crucial role in the formation of the aberrant αSyn species, similar to those found in MSA brains [14,18]. Importantly, others and we have shown that both pharmacologic and genetic inhibition of autophagy leads to the aggregation of endogenous and of exogenously added αSyn in oligodendroglial cells [17,18]. In contrast, Fellner and colleagues reported that inhibiting macroautophagy in the MO3.13 glial cell line—either through bafilomycin A1 treatment or LC3B knockdown—did not substantially alter the levels of αSyn taken up by oligodendrocytes after exposure to soluble or fibrillar protein forms [70]. These discrepancies may reflect differences in the cellular models employed, the doses and conformations of αSyn applied, or whether autophagy was assessed at the level of dynamic flux versus steady-state protein accumulation.

Autophagy implication in the elimination of αSyn aggregates was further highlighted in a recent study where administration of KYP-2047, a prolyl oligopeptidase inhibitor that increases the activity of PP2A phosphatase, resulted in autophagy induction and subsequent reduction in αSyn+ inclusions, improvement of cell viability, and mitigation of pSer129-αSyn levels in an oligodendroglial cell line (OLN) overexpressing human αSyn (OLN-AS7) [57] (Table 2). The authors suggested that the autophagy-modulating effect of prolyl oligopeptidase is a potent disease-modifying mechanism against the development of various α-Synucleinopathies [57].

Rapamycin, an mTOR-dependent pharmacological inducer of macroautophagy, has been previously used for the removal of αSyn aggregates in various PD models [80,81]. In our recently published work, we have proposed that the addition of rapamycin to PFF-treated oligodendrocytes (both cell lines and primary cultures) led to the elimination of the pathological αSyn species formed, setting macroautophagy as a potential target for MSA treatment [17]. Moreover, rapamycin has been administered to patients in various clinical trials for the treatment of a variety of cancers and malignancies, as well as of neurodegenerative diseases [82,83]. However, when rapamycin was delivered to MSA patients for 48 weeks [58] in a double blind, placebo-controlled clinical trial, the patients did not display any clinical improvement, as estimated by the Unified MSA Rating Scale (UMSARS) [58]. This lack of clinical benefit may be attributed to several factors, including limited CNS penetration, suboptimal dosing regimens, insufficient target engagement, delayed initiation of treatment relative to disease progression, and the insensitivity of clinical outcome measures to subtle biological effects.

### 4.2. A Role of CMA in Oligodendroglial αSyn Clearance

CMA is another lysosomal proteolytic pathway that plays a crucial role in the clearance of neuronal αSyn [23,84,85]. Regarding the implication of CMA in MSA, the first indication came from a study that showed immunoreactivity of Hsc70 in GCIs, a chaperone protein required for the proper functioning of the CMA machinery (Figure 3H) [86]. Kawamoto and colleagues proposed that during MSA development, Hsc70 is not able to effectively deliver αSyn at the level of the lysosomal membrane for its binding to the LAMP2A receptor, but instead, CMA is blocked and macroautophagy becomes activated in order to compensate for this defect [86]. On the other hand, according to our recent findings using the OLN cell lines and mouse primary oligodendrocytes, CMA seems to be able to clear both physiological and pathological forms of oligodendroglial αSyn, and interestingly, in cells overexpressing TPPP/p25α (OLN-p25α) treated with αSyn PFFs, macroautophagic activity is impaired and CMA becomes the main autophagic machinery responsible for the removal of pathological αSyn forms [17].

## 5. A Role of the Proteasome in MSA Pathogenesis

### 5.1. Proteasomal Degradation of Oligodendroglial αSyn

Apart from the ALP, the UPS is another major cellular degradation machinery, responsible for the clearance of various ubiquitin-tagged proteins and protein aggregates, amongst which is αSyn [87]. In a seminal study of Mori et al. in 2005, the ubiquitin-like protein NEDD8 (neural precursor cell expressed developmentally downregulated 8 protein) was found to be involved in the generation of ubiquitinated aggregates in the α-Synucleinopathy brain, not only of PD and DLB cases, but also of MSA patients, thus suggesting that the UPS may also exert a significant role in the formation of αSyn-positive aggregates in α-Synucleinopathies in general [88].

Another constituent of GCIs is the ubiquitin carboxyl terminal hydrolase L1 (UCH-L1) that belongs to the deubiquitinase (DUB) family of enzymes, responsible for the removal of ubiquitin that normally promotes proteasomal degradation [89]. It is localized in presynaptic terminals where it interacts with αSyn, thus participating in PD pathogenesis [90,91,92]. Interestingly, it has been shown that UCH-L1 is expressed by oligodendrocytes, being responsible for their architecture via its interaction with the microtubule network under physiological conditions, whereas it is present in the brains of MSA patients in cases of pathology. With respect to its role in autophagy, evidence from oligodendroglial models indicates that UCH-L1 inhibition activates autophagic flux, as shown using a GFP-LC3 reporter in OLN cells and primary oligodendrocytes, which in turn reduces αSyn aggregation [59] (Table 2). By contrast, neuronal studies link UCH-L1 to αSyn accumulation and vulnerability, but without direct flux measurements in oligodendrocytes [90,91,92].

A potential role of the proteasome inhibition in oligodendroglial αSyn accumulation has been additionally proposed in an attempt to study the implication of proteolytic failure in the development of the disease. In particular, Stefanova and colleagues applied systemic proteasome inhibition (5 mg/kg/day for 2 weeks) in PLP-hαSyn mice, a transgenic mouse model of MSA overexpressing human αSyn specifically in oligodendrocytes under the control of the proteolipid protein promoter (PLP) [93]. According to their results, proteasomal impairment resulted in αSyn aggregation and accumulation of high-molecular-weight ubiquitinated proteins within the cytoplasm of oligodendrocytes [93]. Notably, these ubiquitin-positive species represent a broad pool of proteasome substrates rather than αSyn–specific ubiquitination, underscoring that the readouts reflect global proteostatic stress in addition to αSyn-related pathology. As with other systemic proteasome inhibitors, potential off-target toxicity and widespread proteostasis disturbance cannot be excluded, since inhibition was not cell-type specific.

### 5.2. UPS-Mediated Regulation of TPPP/p25α

To date, few studies have focused on the involvement of the proteasome in MSA pathogenesis, with most of them being oriented towards the role of UPS in the clearance of the other main GCI component, TPPP/p25α [94,95]. In the first study, pharmacological inhibition of the proteasome with MG132 (1 μM for 2 h) in TPPP/p25α-transfected HeLa or NRK mammalian cells resulted in increased TPPP/p25α protein levels, suggesting that the protein is mainly degraded via the UPS [94]. Similar results were documented by the same group using CHO10 cells treated with MG132 (10 μM) for 3 h [95]. Both sets of experiments were based on short-term pharmacological inhibition in cell culture, where MG132 can also have off-target effects on other proteases. These results are in agreement with our recent work, where TPPP/p25α was found to be degraded via both the UPS (treatment with *Psmb5* siRNAs, 10 mM, for 72 h) and CMA under pathological conditions [17]. In particular, the contribution of CMA to TPPP/p25α protein degradation was verified by the identification of the KKRFK pentapeptide motif that meets the criteria for a KFERQ-like motif in the TPPP/p25α amino acid sequence and its efficient degradation by the in vitro system of isolated rat brain lysosomes [17]. More recently, it has also been reported that TPPP/p25α is a 20S proteasome substrate and that TPPP/p25α-induced αSyn aggregation resulted in proteasomal dysfunction, both in vitro and in cells, and evoked significant cell cytotoxicity. Treatment with novel small-molecule 20S proteasome enhancers (10 μM for 3 h) prevented TPPP/p25α-induced αSyn fibrillization, proteasome impairment, and cell cytotoxicity, indicating a potential new therapeutic strategy to treat TPPP/p25α/αSyn-associated pathologies, such as those found in MSA [96].

## 6. Conclusions—Future Perspectives

There is distinct promise in the field of neurodegenerative diseases for therapeutics that aim to enhance protein degradation systems, so as to remove toxic aggregated protein species. The current literature has firmly established that the ALP is impaired in MSA, with autophagy-related proteins (e.g., Beclin-1, LC3, LAMP2A, SQSTM1/p62) found to be dysregulated in MSA-like conditions, mitochondrial damage, often irreversible in MSA, worsened by insufficient mitophagy and autophagy-related components co-localizing with αSyn and TPPP/p25α within GCIs, suggesting an overwhelmed degradation system rather than a complete absence of response. This reveals a “failed cellular response” model: the cell attempts to clear aggregates, but progressive autophagy dysfunction renders this effort inadequate, leading to protein accumulation, neuroinflammation, and subsequent widespread degeneration.

The dysregulation of autophagy in MSA is not merely a consequence of oligodendroglial dysfunction and neuronal degeneration, but a potential driver of disease progression. Modulating autophagy represents one of the most promising therapeutic frontiers in MSA, offering a path to not just symptom management but disease modification. Others and we have reported that ALP enhancement might open new therapeutic opportunities for slowing down the degenerative process in patients with α-Synucleinopathies. However, therapeutic success depends, amongst others, on early and accurate diagnosis, targeted and safe modulation of autophagy, and integration of autophagy-targeting agents into personalized treatment regimens. While modulation of autophagy represents an attractive therapeutic frontier, clinical experience underscores the challenges. For example, the sirolimus (rapamycin) futility trial in MSA failed to show clinical improvement despite preclinical promise [58] (Table 2). This negative result highlights issues of CNS penetration, dosing, target engagement, timing of intervention, and the sensitivity of clinical outcomes.

The literature on autophagy in MSA is inherently complex, reflecting overlapping degradation pathways and context-dependent findings. Proteins such as αSyn, TPPP/p25α, and autophagy regulators act through partially convergent mechanisms, making interpretation of the data obtained rather challenging. We acknowledge this as one of the “current challenges” highlighted in our abstract: the need to disentangle overlapping proteostatic mechanisms while retaining a coherent framework for therapeutic development. Explicitly recognizing this complexity emphasizes why careful experimental design, better disease models, and rigorous biomarker strategies are near-term priorities.

With advances in molecular tools and drug delivery platforms, autophagy-based therapies may soon move from theoretical promise to clinical practice in MSA. Such advancements may include:Combinatorial therapies combining autophagy enhancers with anti-inflammatory agents or neuroprotectants (e.g., antioxidants or mitochondrial function boosters).Use of patient-derived biological fluids (blood, serum, plasma) to identify peripheral biomarkers of autophagy dysfunction that may mirror brain pathology in MSA.Application of patient iPSC-derived oligodendrocytes and 3D organoids to investigate autophagy-related mechanisms in MSA development.Screening of autophagy-modulating compounds in these humanized models to uncover novel ALP-targeted therapeutic opportunities.

## Figures and Tables

**Figure 1 ijms-26-10204-f001:**
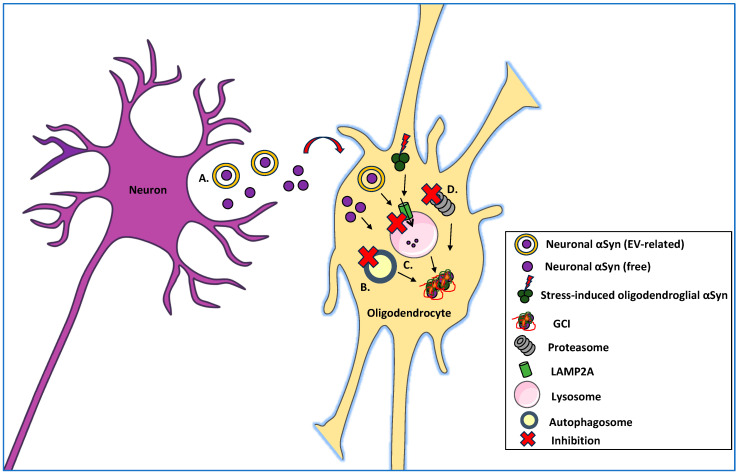
The origin of oligodendroglial αSyn. (**A**) Neuronal αSyn is released either free or associated with extracellular vesicles and subsequently internalized by oligodendrocytes. (**B**–**D**) Under conditions of impaired proteostasis, stress-induced endogenous oligodendroglial αSyn expression is increased, and its degradation via (**B**) macroautophagy, (**C**) chaperone-mediated autophagy (CMA), or (**D**) the ubiquitin–proteasome system (UPS) is compromised. Inhibition (X) of these clearance pathways promotes abnormal αSyn accumulation and formation of protein inclusions within oligodendrocytes.

**Figure 2 ijms-26-10204-f002:**
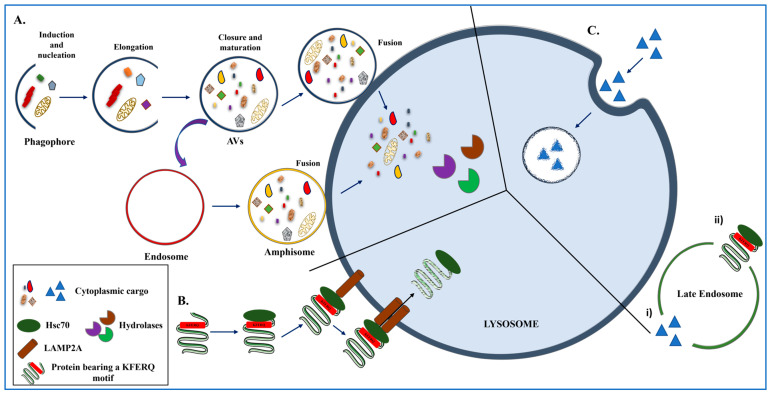
Three distinct types of autophagy operate in mammalian cells. (**A**) Macroautophagy is mediated by the formation of double-membrane vesicles, called autophagosomes (autophagic vacuoles, AVs). The main steps of this degradation procedure include the formation of the phagophore engulfing the cytoplasmic material (induction and nucleation step), followed by the elongation of the phagophore membrane and the fusion of its edges to form the AV (elongation, closure, and maturation step). Finally, the AV is delivered to the lysosome to form the autolysosome (fusion step). The digestive enzymes of the lysosome (hydrolases) are responsible for the degradation of the luminal material. Alternatively, the AV fuses with an endosome to form an amphisome, prior to its fusion with the lysosome. (**B**) CMA is a very selective pathway of autophagy, responsible for the removal of specific proteins bearing the KFERQ (or a biochemically related motif) recognition sequence. These proteins are recognized by the Hsc70 chaperone protein, which then delivers the substrate to the multimerized LAMP2A receptor on the lysosome for degradation. (**C**) Microautophagy refers to the direct engulfment of the cytoplasmic cargo (proteins, cell components) to lysosomes or late endosomes via membrane invagination. It can either be in bulk (**i**) or selective (**ii**) through the targeting of cytosolic proteins by Hsc70 through eMI.

**Figure 3 ijms-26-10204-f003:**
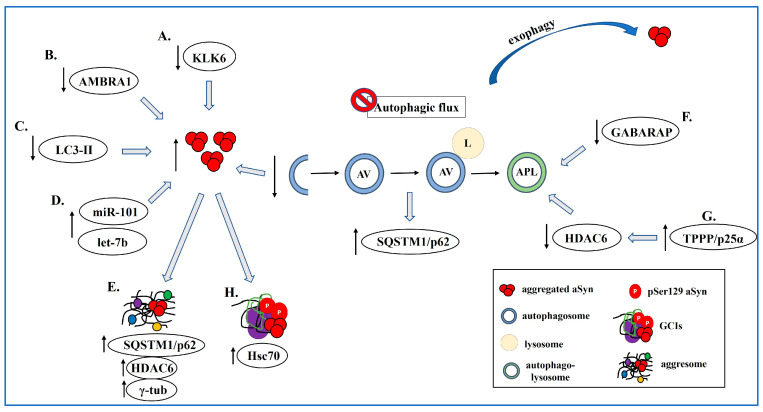
Proposed mechanisms of ALP impairment leading to MSA pathogenesis. (**A**) Downregulation of the serine protease kallikrein-6 (KLK6), which is responsible for the clearance of αSyn, leads to protein accumulation in disease states. (**B**) Silencing of the autophagy/beclin1 regulator 1 (AMBRA1) function, an upstream protein of autophagy, causes the formation of αSyn aggregates. (**C**) Decreased levels of LC3-II demonstrate autophagy dysregulation that finally leads to increased αSyn aggregation. (**D**) Up-regulated levels of the miR-101 and let-7b microRNAs, specifically suppressing autophagy-related genes, are found in the brain of MSA patients and lead to the formation of aberrant αSyn species, due to defective autophagy. (**E**) The accumulation of SQSTM1/p62, HDAC6, and gamma-tubulin (γ-tub), along with αSyn and other protein aggregates, into a structure called an aggresome, suggests that it represents a special cellular protective response to proteotoxic stress. (**F**) In MSA brains, the reduced levels of the ubiquitin-like modifier GABARAP, which plays a crucial role in autophagosome maturation, reveal impaired autophagic flux and elevated levels of adaptor proteins, such as SQSTM1/p62. (**G**) Elevated levels of the oligodendroglial-specific phosphoprotein TPPP/p25α inhibit the activity of histone deacetylase 6 (HDAC6), finally leading to impaired autophagic flux, a fact that induces the secretion of αSyn aggregates via exophagy, an unconventional autophagic secretory process. (**H**) Hsc70, a crucial player in CMA, shows strong immunoreactivity in GCIs, thus indicating the implication of this autophagic pathway in MSA.

**Table 1 ijms-26-10204-t001:** Differential regulation of proteostasis checkpoints between oligodendrocytes and microglial cells in MSA (+: positive, -: negative, ↑: increase, ↓: decrease).

Checkpoint/Marker	Oligodendrocytes		Microglia
	Aggresomes	GCIs	
p62/SQSTM1	+	+	
HDAC6	+	-	
γ-tubulin	+	+	
NBRI	-	+	
LC3-II	**↑**		
Lysosomal status	**↓** GABARAP (maturation)		**↑** activation
αSyn	**↑** aggregation		**↑** uptake and clearance

## Data Availability

No new data were created or analyzed in this study. Data sharing is not applicable to this article.

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
