# Peer review of "Autophagy–Lysosome Pathway in Multiple System Atrophy Pathogenesis: The Best Is Yet to Come"

_ijms, 2025, doi:10.3390/ijms262010204_

Round 1

Reviewer 1 Report

Comments and Suggestions for Authors

Multiple Systems Atrophy (MSA) is a neurodegenerative disease linked to α-synuclein accumulation in oligodendrocytes, however, its etiology is not well understood. Evidence points at impairment of the autophagy-lysosome pathway (ALP) and proteosome (UPS), however the exact mechanisms remain unclear. The lack of clarity on the early mechanisms of MSA pathology makes it challenging to diagnose and treat making this review very relevant to the field. Here, Mavroeidi and Xilouri frame the current evidence of ALP impairment into perspective, proposing therapeutic targets and highlighting areas where further research is needed.

Overall, the manuscript is well written, and the grammar and sentence structure are good. However, some transitions and paragraph organization could be improved to better guide the reader through the focus of the discussion. Below, I outline specific comments to improve clarity and flow.

  1. Section 1: Introduction
  • In lines 38-40 the authors raise an important knowledge gap regarding the origin of aSyn found in oligodendrocytes in MSA. In lines 41-45, they briefly introduce an “exogenous protein model” of prion-like spread of aSyn released from neurons but then close the paragraph without fully developing it. The following paragraph (line 46) introduces an “endogenous protein model”, supported by their work and that of others, though the transition between the text in lines 45 and 46 could be smoother. Finally, in lines 50-51 they introduce the autophagy-lysosome pathway (ALP) which is at the center of this review. As currently written, ALP is presented mainly in connection to the “endogenous protein model” even though at least two of the references cited for the prion-like spread also implicate it (e.g., references 11 and 12). To improve clarity the authors could strengthen the transitions between these sections and explicitly note that ALP is relevant to both exogenous and endogenous origin models. This would provide the readers with a more integrated view of how these proposed mechanisms relate to one another.

  • In lines 54-56 the authors mention synucleinopathies associated with ALP failure, while leaving out MSA. Since the subsequent text discusses evidence of ALP failure, including their own work, it would strengthen this manuscript to cite this directly alongside the other synucleinopathies. This is also discussed in reference #68 (Fellner, et al 2021).

  1. Figure 1: It would be helpful if the text in Section 2 pointed readers to the specific parts of the model (e.g., A, B, C) especially since the order of the mechanisms described in the text is different from the figure. To make figure 1 easier to interpret at a glance, it would be helpful to label the mechanisms directly on the corresponding diagrams.

  1. Section 3: This section is very dense, and the current organization makes its focus hard to follow. In lines 128-129, the authors state that they will focus primarily on the role of autophagy and, to a lesser extent, UPS. However, the way this section is organized makes it difficult to distinguish when the discussion is centered on one versus the other. This may reflect the known crosstalk between these mechanisms (see: Limanaqi, Fiona, et al. Int. J. Mol. Sci. 2020), but clearer subheading could help readers follow the intended emphasis.

  1. Sections 3-5 provide a comprehensive overview of the current knowledge of autophagy and proteasome impairments, putting potential therapeutic targets into perspective for MSA. However, the way the material is presented makes the focus difficult to follow. Clearer framing and the use of subheadings could help guide readers through the main points, and these sections would benefit from being divided into subsections, for example, highlighting the different proteins.

  1. The discussion is sometimes difficult to follow, though I think it reflects the genuine complexity of literature. The authors have done a good job of covering the relevant studies, but the proteins discussed here have overlapping mechanisms which can make the story challenging to explain. To avoid confusion for readers, additional framing or clarification would be helpful. It may help to acknowledge this directly and frame it as one of the “current challenges” noted in the abstract line 23, which would further emphasize the relevance of this review.

  1. Figure 2 provides a clear visual of the proteins and mechanisms described in sections 3-5 and how they relate to each other in the context of autophagy, UPS and MSA pathogenesis. However, it is not explicitly linked to the main text. Referring to this figure in the relevant sections would help integrate it more fully and guide reader in connecting the visual with the discussion.

Author Response

Reviewer 1

Multiple Systems Atrophy (MSA) is a neurodegenerative disease linked to α-synuclein accumulation in oligodendrocytes, however, its etiology is not well understood. Evidence points at impairment of the autophagy-lysosome pathway (ALP) and proteosome (UPS), however the exact mechanisms remain unclear. The lack of clarity on the early mechanisms of MSA pathology makes it challenging to diagnose and treat making this review very relevant to the field. Here, Mavroeidi and Xilouri frame the current evidence of ALP impairment into perspective, proposing therapeutic targets and highlighting areas where further research is needed.

Overall, the manuscript is well written, and the grammar and sentence structure are good. However, some transitions and paragraph organization could be improved to better guide the reader through the focus of the discussion. Below, I outline specific comments to improve clarity and flow.

Response: We greatly appreciate the reviewer’s comments for our manuscript.

  1. Section 1: Introduction
  • In lines 38-40 the authors raise an important knowledge gap regarding the origin of aSyn found in oligodendrocytes in MSA. In lines 41-45, they briefly introduce an “exogenous protein model” of prion-like spread of aSyn released from neurons but then close the paragraph without fully developing it. The following paragraph (line 46) introduces an “endogenous protein model”, supported by their work and that of others, though the transition between the text in lines 45 and 46 could be smoother. Finally, in lines 50-51 they introduce the autophagy-lysosome pathway (ALP) which is at the center of this review. As currently written, ALP is presented mainly in connection to the “endogenous protein model” even though at least two of the references cited for the prion-like spread also implicate it (e.g., references 11 and 12). To improve clarity the authors could strengthen the transitions between these sections and explicitly note that ALP is relevant to both exogenous and endogenous origin models. This would provide the readers with a more integrated view of how these proposed mechanisms relate to one another.
  • In lines 54-56 the authors mention synucleinopathies associated with ALP failure, while leaving out MSA. Since the subsequent text discusses evidence of ALP failure, including their own work, it would strengthen this manuscript to cite this directly alongside the other synucleinopathies. This is also discussed in reference #68 (Fellner, et al 2021).

Response: In accordance with the reviewer’s valuable suggestion, we have carefully revised the transitions between sections throughout the manuscript to ensure a more seamless flow and to provide readers with a comprehensive and integrated understanding of how the proposed mechanisms interrelate.

  1. Figure 1:It would be helpful if the text in Section 2 pointed readers to the specific parts of the model (e.g., A, B, C) especially since the order of the mechanisms described in the text is different from the figure. To make figure 1 easier to interpret at a glance, it would be helpful to label the mechanisms directly on the corresponding diagrams.

Response: Following the reviewer’s suggestion, we have revised Figure 1 (now presented as Figure 2 in the revised manuscript) by labeling panels A, B, and C to clearly indicate the different pathways operating in the autophagy–lysosomal pathway (ALP).

  1. Section 3: This section is very dense, and the current organization makes its focus hard to follow. In lines 128-129, the authors state that they will focus primarily on the role of autophagy and, to a lesser extent, UPS. However, the way this section is organized makes it difficult to distinguish when the discussion is centered on one versus the other. This may reflect the known crosstalk between these mechanisms (see: Limanaqi, Fiona, et al. Int. J. Mol. Sci. 2020), but clearer subheading could help readers follow the intended emphasis.

Response: Following the reviewer’s suggestion, we have reorganized this section and incorporated subheadings to enhance clarity and improve the readability of the mechanisms discussed.

  1. Sections 3-5provide a comprehensive overview of the current knowledge of autophagy and proteasome impairments, putting potential therapeutic targets into perspective for MSA. However, the way the material is presented makes the focus difficult to follow. Clearer framing and the use of subheadings could help guide readers through the main points, and these sections would benefit from being divided into subsections, for example, highlighting the different proteins.

Response: Following the reviewer’s suggestion, we have reorganized this section and incorporated subheadings to enhance clarity and improve the readability of the mechanisms discussed.

  1. The discussion is sometimes difficult to follow, though I think it reflects the genuine complexity of literature. The authors have done a good job of covering the relevant studies, but the proteins discussed here have overlapping mechanisms, which can make the story challenging to explain. To avoid confusion for readers, additional framing or clarification would be helpful. It may help to acknowledge this directly and frame it as one of the “current challenges” noted in the abstract line 23, which would further emphasize the relevance of this review.

Response: In response to the reviewer’s suggestion, we have revised the discussion to emphasize the intricate nature of the mechanisms involved, while also addressing the prevailing challenges and recent progress in this area of research.

  1. Figure 2provides a clear visual of the proteins and mechanisms described in sections 3-5 and how they relate to each other in the context of autophagy, UPS and MSA pathogenesis. However, it is not explicitly linked to the main text. Referring to this figure in the relevant sections would help integrate it more fully and guide reader in connecting the visual with the discussion.

Response: We are grateful to the reviewer for this helpful suggestion. Accordingly, we have revised the manuscript to explicitly reference this figure (now presented as Figure 3 in the revised manuscript) within the relevant sections of the text, thereby enhancing the integration between the visual content and the discussion, and improving overall manuscript coherence.

Reviewer 2 Report

Comments and Suggestions for Authors

Reviewer Comments
This is a comprehensive and well-written review that provides extensive knowledge on autophagy and α-synuclein pathology in MSA by Mavroeidi and Xilouri. The manuscript is clearly organized, appropriately referenced, and accessible to a broad audience interested in neurodegeneration. To further strengthen this article, I suggest sharpening the conceptual framing in the introduction, standardizing terminology and acronyms throughout, and adding brief signposts to reconcile divergent findings across models and cell types. A few flow edits (e.g., tightening section transitions, adding a short roadmap, and a summary table/figure) would also improve readability. Please find the detailed, line-by-line suggestions below. 
1.    Line 8: Please capitalize “l” to “L.”
2.    Line 29: I suggest replacing “orphan” with “rare”. 
3.    Line 69: modify “CGI” to “GCI”. 
4.    The abstract says MSA data are “scant,” yet the first pages cite multiple MSA-specific tissue/cell findings. Please replace “scant” with “emerging but less extensive than PD.”
5.    To avoid perceived contradiction, please clarify α-syn. Add a one-sentence bridge contrasting neuronal uptake (free or EV-associated) with stress-induced endogenous expression in oligodendrocytes. (You already state low baseline expression and EV/internalization. Also note increases when ALP/UPS are impaired.)
6.    Reviews are understood better with visuals; please include an introductory schematic showing (i) neuronal α-syn release → oligodendroglial uptake (free/EV) and (ii) stress-induced endogenous α-syn, both converging on GCI formation, with ALP/CMA/UPS checkpoints annotated.
7.    Lines 79–80: When you write “endosomal microautophagy,” define it on first use as eMI and keep that acronym thereafter.
8.    Scope signpost & wording (Lines 120–129): Please add a one-sentence roadmap (“Here we synthesize ALP/UPS evidence in MSA, noting contrasts with PD”) and briefly clarify ‘primary oligodendrogliopathy/secondary neuropathy’ for non-specialists.
9.    Neurosin/KLK6 paragraph (Lines 130–136): Specify whether the reported changes are mRNA, protein, or activity, and include brain region and n where possible.
10.    Figure nudge (Section 3 overall): A small schematic showing aggresome/ALP checkpoints (p62, HDAC6, γ-tubulin) and where microglia vs oligodendrocytes show changes would make this section easier to understand.
11.    Please reconcile with the earlier negative result. You state that pharmacologic/genetic autophagy inhibition increases α-syn aggregation in oligodendroglia; please briefly reconcile this with the earlier Fellner study, where macroautophagy blockade (bafilomycin/LC3B KD) did not drive GCI-like build-up, noting model/dose/time or readouts (flux vs steady state). (Lines 275–281; Lines 251–260).
12.    Bench-to-bedside gap for rapamycin. You nicely cite rapamycin’s in vitro clearance, but a futility clinical trial in MSA. Please add 1–2 sentences on likely reasons (CNS exposure, dosing, target engagement, timing, outcome sensitivity) to contextualize why macroautophagy induction may fail clinically. (Lines 295–305).
13.    Consider a compact table listing each ALP-targeting strategy (e.g., KYP-2047 → PP2A/↑autophagy; rapamycin → mTOR/↑macroautophagy), model system, outcomes (LC3-II, p62, pSer129-αSyn), and translational status to give readers a quick comparative view.
14.    Lines 328–336 (UCH-L1 paragraph): Fix grammar (“can acts” to “can act”). Also, clarify evidence and cell type: is UCH-L1’s role as an autophagy modulator shown with flux assays (e.g., LC3-II turnover/p62 with lysosomal blockade) in oligodendrocytes, or inferred from neuronal/presynaptic data?
15.    Lines 339–344 & 348–353 (proteasome inhibition studies): Please add dose/duration and note potential off-target toxicity for systemic proteasome inhibition in mice and cells. Specify whether the readouts were α-syn–specific versus global ubiquitinated proteins.
16.    Conclusion: Clinical translation caveat. The paragraph is optimistic about autophagy-based therapies and combinations; please consider explicitly acknowledging the negative rapamycin trial in MSA and briefly stating what makes next-gen strategies different (e.g., CMA-targeted, cell-type–specific, biomarker-guided). A short bullet list of near-term priorities would sharpen the “Future Perspectives.”

Author Response

Reviewer 2

This is a comprehensive and well-written review that provides extensive knowledge on autophagy and α-synuclein pathology in MSA by Mavroeidi and Xilouri. The manuscript is clearly organized, appropriately referenced, and accessible to a broad audience interested in neurodegeneration. To further strengthen this article, I suggest sharpening the conceptual framing in the introduction, standardizing terminology and acronyms throughout, and adding brief signposts to reconcile divergent findings across models and cell types.

Response: We greatly appreciate the reviewer’s comments for our manusript.

A few flow edits (e.g., tightening section transitions, adding a short roadmap, and a summary table/figure) would also improve readability. Please find the detailed, line-by-line suggestions below. 

  1. Line 8: Please capitalize “l” to “L.”
  2.   Line 29: I suggest replacing “orphan” with “rare”. 
  3.  Line 69: modify “CGI” to “GCI”. 
  4. The abstract says MSA data are “scant,” yet the first pages cite multiple MSA-specific tissue/cell findings. Please replace “scant” with “emerging but less extensive than PD.”

Response: We thank the reviewer for the constructive feedback. Comments 1–4 have been thoroughly addressed and the corresponding revisions have been incorporated into the relevant sections of the manuscript.

  1.    To avoid perceived contradiction, please clarify α-syn. Add a one-sentence bridge contrasting neuronal uptake (free or EV-associated) with stress-induced endogenous expression in oligodendrocytes. (You already state low baseline expression and EV/internalization. Also note increases when ALP/UPS are impaired.)

Response: Following the reviewer’s suggestion, we have inserted the following sentence bridgeAlternatively, αSyn in oligodendrocytes may not only reflect neuronal uptake but also stress-induced endogenous expression, particularly under conditions where proteolytic systems, such as the autophagy–lysosome pathway (ALP) or the ubiquitin–proteasome system (UPS), are impaired (Figure 1)”.

  1.    Reviews are understood better with visuals; please include an introductory schematic showing (i) neuronal α-syn release → oligodendroglial uptake (free/EV) and (ii) stress-induced endogenous α-syn, both converging on GCI formation, with ALP/CMA/UPS checkpoints annotated.

Response: In response to the reviewer’s suggestion we provide a schematic regarding the possible mechanisms that underlie αSyn accumulation within oligodendrocytes in MSA (now presented as Figure 1 in the revised manuscript).

7.    Lines 79–80: When you write “endosomal microautophagy,” define it on first use as eMI and keep that acronym thereafter.

Response: This is corrected.

  1. Scope signpost & wording (Lines 120–129): Please add a one-sentence roadmap (“Here we synthesize ALP/UPS evidence in MSA, noting contrasts with PD”) and briefly clarify ‘primary oligodendrogliopathy/secondary neuropathy’ for non-specialists.

Response: Following the reviewer’s suggestion, we have included the one-sentence roadmap and clarified further the “primary oligodendrogliopathy/secondary neuropathy” nature of MSA.

  1. Neurosin/KLK6 paragraph (Lines 130–136): Specify whether the reported changes are mRNA, protein, or activity, and include brain region and n where possible.

Response: In response to the reviewer’s suggestion, we have made the respective additions.

  1. Figure nudge (Section 3 overall): A small schematic showing aggresome/ALP checkpoints (p62, HDAC6, γ-tubulin) and where microglia vs oligodendrocytes show changes would make this section easier to understand.

Response: Following the reviewer’s valuable suggestion, we now provide Table 1 depicting the differential regulation of proteostasis chackpoints between oligodendrocytes and microglial cells in MSA (section 3.4).

  1. Please reconcile with the earlier negative result. You state that pharmacologic/genetic autophagy inhibition increases α-syn aggregation in oligodendroglia; please briefly reconcile this with the earlier Fellner study, where macroautophagy blockade (bafilomycin/LC3B KD) did not drive GCI-like build-up, noting model/dose/time or readouts (flux vs steady state). (Lines 275–281; Lines 251–260).

Response: In response to the reviewer’s suggestion, the contradictory results of the outcome of autophagy inhibition to αSyn accumulation/aggregation in the context of MSA have been reconciled in the revised manuscript (section 4.1, first paragraph).

  1. Bench-to-bedside gap for rapamycin. You nicely cite rapamycin’s in vitro clearance, but a futility clinical trial in MSA. Please add 1–2 sentences on likely reasons (CNS exposure, dosing, target engagement, timing, outcome sensitivity) to contextualize why macroautophagy induction may fail clinically. (Lines 295–305).

Response: Following the reviewer’s suggestion, we have included one sentence providing plausible explanation for the ineffectiveness of the rapamycin clinical trial in MSA. “This lack of clinical benefit may be attributed to several factors, including limited central nervous system penetration, suboptimal dosing regimens, insufficient target engagement, delayed initiation of treatment relative to disease progression, and the insensitivity of clinical outcome measures to subtle biological effects” (section 4.1, third paragraph).

  1.    Consider a compact table listing each ALP-targeting strategy (e.g., KYP-2047 → PP2A/↑autophagy; rapamycin → mTOR/↑macroautophagy), model system, outcomes (LC3-II, p62, pSer129-αSyn), and translational status to give readers a quick comparative view.

Response: According to the reviewer’s suggestion, we now provide Table 2 depicting experimental and clinical evidence from autophagy–lysosome–proteostasis–targeting strategies in MSA (section 4.2).

  1.  Lines 328–336 (UCH-L1 paragraph): Fix grammar (“can acts” to “can act”). Also, clarify evidence and cell type: is UCH-L1’s role as an autophagy modulator shown with flux assays (e.g., LC3-II turnover/p62 with lysosomal blockade) in oligodendrocytes, or inferred from neuronal/presynaptic data?

Response: In response to the reviewer’s suggestion, all respective grammar errors have been corrected.

  1. Lines 339–344 & 348–353 (proteasome inhibition studies): Please add dose/duration and note potential off-target toxicity for systemic proteasome inhibition in mice and cells. Specify whether the readouts were α-syn–specific versus global ubiquitinated proteins.

Response: Following the reviewer’s suggestion, the respective information is included in the revised manuscript (section 5.1).

  1. Conclusion: Clinical translation caveat. The paragraph is optimistic about autophagy-based therapies and combinations; please consider explicitly acknowledging the negative rapamycin trial in MSA and briefly stating what makes next-gen strategies different (e.g., CMA-targeted, cell-type–specific, biomarker-guided). A short bullet list of near-term priorities would sharpen the “Future Perspectives.”

Response: According to the reviewer’s suggestion, we have modified our discussion accordingly by acknowledging the existing clinical translation caveat of using autophagy-based therapies in MSA and proposing potential advancements in molecular tools and drug delivery platforms.

Round 2

Reviewer 1 Report

Comments and Suggestions for Authors

I appreciate the careful attention to the points I raised in my initial review. The authors have addressed my feedback on the focus and organization of the introduction and the section headers. These revisions have improved readability and logical flow and have strengthened the manuscript significantly. I particularly like the inclusion of the new Figure 1 and tables, which help clarify the key points and guide the reader through the text.

I have some minor suggestions to further strengthen this work:

Table 2: It would be helpful to include an additional column with the relevant references to better connect the summarized information with the supporting literature.

Author Response

Response: We thank the reviewer for the positive feedback. In response to the reviewer's comment, we have modified Table 2 in the second revision of our manuscript (ijms-3851405 R2), including an additional column with the relevant references to better connect the summarized information with the supporting literature.